# Investigation on Hydroplaning Behaviors of a Patterned Tire on a Steel Bridge Deck Pavement

**Yang Liu** *, **Zhendong Qian, Changbo Liu** and **Qibo Huang**

Intelligent Transportation System Research, Southeast University, Nanjing 211189, China;
qianzd@seu.edu.cn (Z.Q.); liuchangbo@seu.edu.cn (C.L.); huangqibo@seu.edu.cn (Q.H.)
* Correspondence: seuliuyang@seu.edu.cn

**Abstract:** The hydroplaning propensity on the steel bridge deck pavement (SBDP) is higher than ordinary road pavements. In this study, the objective is to develop a hydroplaning model to evaluate the hydroplaning behaviors for SBDPs. To achieve this goal, a finite element (FE) model of a 3D-patterned radial tire model was developed at first, and the grounding characteristics of tire on the SBDP were calculated as an initial condition for the follow-up hydroplaning analysis. The X-ray CT scanning device and Ostu thresholding method were used for image processing of pavement surface topography, and the 3D FE model of SBDP was established by the reverse stereological theory and voxel modeling technique, which can accurately reconstruct the pavement morphology. A fluid model was established to simulate the dynamic characteristics of water film between the tire and SBDP. On this basis, the tire–fluid–pavement interaction model was developed based on the CEL (Couple Eulerian–Lagrangian) algorithm, and it was verified by the hydroplaning empirical equations. Finally, the hydroplaning behaviors on the SBDP were studied. The findings from this study can provide a tool for hydroplaning evaluation on SBDPs, and will be helpful to improve the driving safety of SBDP in rainy days.

**Keywords:** hydroplaning behavior; patterned tire; steel bridge deck pavement; tire–fluid–pavement interaction model; numerical simulation





## 1. Introduction

Hydroplaning on pavement in rainy conditions is a common phenomenon threatening driving safety [1]. When a vehicle runs on a wet pavement, the water between the tire and pavement flows under the contact force at the tire–pavement interface. Some water is drained through the channels formed by the tire pattern and pavement surface macrotexture, while the undrained water can generate the hydrodynamic pressure to the tire, which can reduce the contact between the tire and pavement. Once the vehicle reaches a critical speed, the tire floats entirely on the water and vehicle instability occurs [1,2]. This phenomenon is called hydroplaning and the critical speed is called the hydroplaning speed. Hydroplaning is the main cause of accidents in rainy days, and reducing the risk of hydroplaning is of great importance to driving safety.

Hydroplaning behaviors on an asphalt pavement are related to the tire characteristics, pavement surface topography, and water film on the pavement surface. The early research on hydroplaning behaviors was carried out by experimental methods. Horne et al. [1] studied the effect of the tire inflation pressure on the hydroplaning speed. The studies contributed to the development of the well-known NASA (National Aeronautics Space Administration, Washington, DC, USA) hydroplaning equation, as shown in the Equation (1) which is still widely used as of today.

$$
\begin{cases}
V_{h,rolling} = 6.36\sqrt{P} \\
V_{h,locked} = 5.43\sqrt{P}
\end{cases}
\tag{1}
$$

where $V_{h,rolling}$ and $V_{h,locked}$ are the hydroplaning speeds for rolling tire and locked tire, respectively, [km/h], and $P$ is the inflation pressure, [kPa]. The NASA hydroplaning equation can be used to obtain the hydroplaning speeds for smooth tires with closed pattern treads, and for rib-tread tires on fluid-covered runways where the fluid depth exceeds the tire tread depth [2]. However, the effects of the pavement surface topography, water film depth, and some other tire characteristics are not involved in the NASA hydroplaning equation. Gallaway et al. [3] developed a hydroplaning equation based on the hydroplaning test, and the quantitative relationship between tire inflation pressure, tire pattern depth, water film thickness, pavement surface macrotexture characteristic, and hydroplaning speed was determined in the equation. The equation, named the Gallaway hydroplaning equation, is shown in Equation (2).

$$\begin{cases} V_P = (SD)^{0.04}(P_t)^{0.3}(TRD+1)^{0.06}A \\ A = \max\{[\frac{10.409}{t_w^{0.06}} + 3.507], [\frac{28.952}{t_w^{0.06}} - 7.819](MTD)^{0.04}\} \end{cases} \tag{2}$$

where $V_p$ is the hydroplaning speed, [mph]; $SD$ is the slip ratio of tire, [%], which is defined as the ratio of the slip speed to the vehicle speed; $P_t$ is the tire inflation pressure, [psi]; $TRD$ is the tire pattern depth, [1/32 inch]; $t_w$ is the water film thickness, [inch]; $MTD$ is the mean texture depth of pavement surface. However, the hydroplaning test needs a demanding site and equipment, and it has a high test cost and potential safety hazards. Therefore, the experimental studies are not widely applied to the hydroplaning analysis.

In view of the complex dynamics involved in the hydroplaning behaviors, the numerical simulation has been applied to hydroplaning analysis since the 1970s. At present, the finite element (FE) simulation technology based on the Couple Eulerian–Lagrangian (CEL) algorithm has become an important method for hydroplaning analysis. Many researchers conducted numerical studies to reveal the influence of tire pattern characteristics on hydroplaning behaviors [4–6]. The purpose of these studies is for the safety design of tires, but the influence of pavement surface topography is not well modeled. Ong et al. [7,8] developed a tire–fluid–pavement interaction model to evaluate the driving safety on a wet asphalt pavement, and revealed the influences of water film depth and pavement surface texture on the hydroplaning speed. It was found that the hydroplaning speed decreased with the increase in water film thickness, and tended to stabilize when the water film thickness reached 2 mm, and the improvement of pavement microtexture in the 0.2 to 0.5 mm range can delay hydroplaning. However, the tire in the model was smooth tire, and the modeling of the pavement surface topography was rough. With the application of X-ray CT technology to the numerical modeling of pavement meso-structure [9–11], the model accuracy of pavement surface topography improves. Srirangam et al. [12–14] developed a 3D FE model of pavement surface topography based on the X-ray CT technology, which can reconstruct the structure details of pavement surface texture well. On this basis, they established a hydroplaning model for patterned tire, and verified the effectiveness of hydroplaning model by the experimental results. Zhu et al. [15,16] developed a tire–fluid–pavement model based on the CEL algorithm, and it was found that higher tire inflation pressure, thinner water film, and more abundant macrotexture can enhance the hydroplaning speed through numerical analysis.

The hydroplaning analysis is an essential element of driving safety evaluation on asphalt pavement, but the study of hydroplaning behaviors on the steel bridge deck pavement (SBDP) has never been involved. The hydroplaning propensity on SBDP is generally higher than road pavement due to the small pavement macrotexture depth [17,18]. The SBDP is a special pavement structure type, which is a thin-layer structure paved on the orthotropic steel bridge deck [17–19]. Due to the special operating conditions and unique structure features for bridges, the structure design and material application for SBDP are different from conventional road pavement. The previous research [20] showed that the tire deformation and grounding characteristics when the vehicle is running on the SBDP differ from that on road pavement. Accordingly, it is necessary to develop a hydroplaning

model for SBDP. This study aims to investigate the hydroplaning behaviors on the SBDP through establishing a tire–fluid–SBDP interaction model by *ABAQUS* software, in which the patterned tire model and SBDP model with complex macrotexture are involved.

## 2. Modeling of 3D-Patterned Tire

### 2.1. Material Parameter Acquisitions of Tire

In this study, the 225/60R17 radial tire was selected to establish the patterned tire model. The main materials for radial tire can be divided into the rubber material and rubber-cord composite material. The rubber materials are distributed in the tread, cap ply, belt ply, carcass, inside liner, sidewall, and bead filler of tire. The rubber-cord composite materials are distributed in the cap ply, belt ply, carcass, and bead of tire.

#### 2.1.1. Rubber Materials

The hyperelastic property of rubber materials can be characterized by the Yeoh model [21,22], as expressed in Equation (3).

$$W = C_{10}(I_1 - 3) + C_{20}(I_1 - 3)^2 + C_{30}(I_1 - 3)^3 \tag{3}$$

where $W$ is the strain energy per unit volume; $I_1$ is the strain invariant; $C_{i0}$ ($i = 1, 2, 3$) are the material parameters, which can be determined by fitting test data. In this study, the uniaxial tensile test was conducted to acquire the hyperelastic characterization parameters for different rubber materials [23], as shown in Figure 1. Through fitting the test data by the Yeoh model, the values of model parameters for different rubber materials are listed in Table 1.

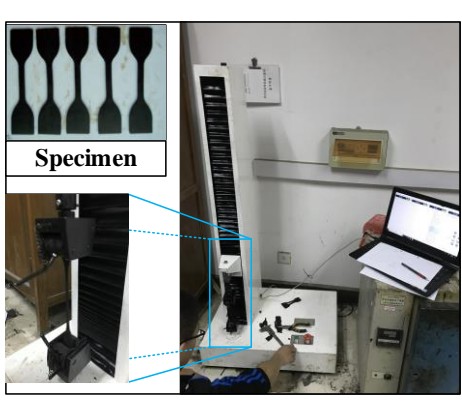

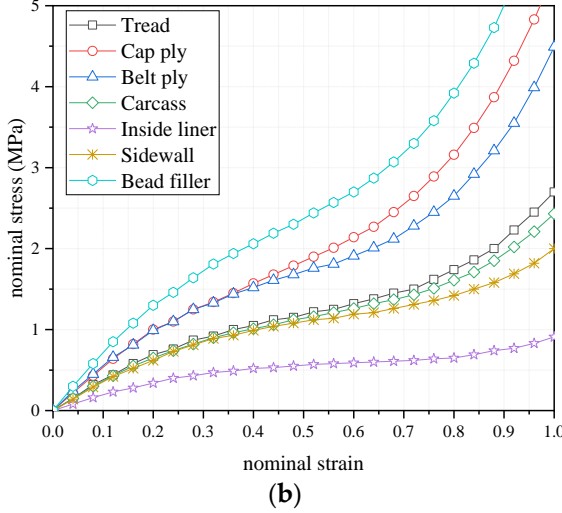

(**a**)　　　　　　　　　　　　　　　　　(**b**)

**Figure 1.** Uniaxial tensile test for different rubber materials (**a**) direct tensile testing machine; (**b**) test results of different rubber materials.

**Table 1.** Yeoh model parameters for different rubber materials.

| Rubber Types | Tread | Cap Ply | Belt Ply | Carcass | Inside Liner | Sidewall | Bead Filler |
|---|---|---|---|---|---|---|---|
| Density (kg/m$^3$) | 1150 | 1148 | 1205 | 1124 | 1254 | 1105 | 1216 |
| C10 | 0.7052 | 0.9950 | 1.0233 | 0.6711 | 0.3586 | 0.6582 | 1.3385 |
| C20 | −0.1802 | −0.2127 | −0.2860 | −0.1640 | −0.1037 | −0.1584 | −0.3187 |
| C30 | 0.0652 | 0.1168 | 0.1168 | 0.0564 | 0.0262 | 0.0448 | 0.1520 |

2.1.2. Rubber-Cord Composite Materials

The rubber-cord composite material is anisotropic. In the radial tire, the cords are embedded into the rubber material at a certain distance and angle, to form the reinforcement layer. In this study, the rubber-cord composite materials were modeled through defining one or more embedded rebar elements in the rubber, as shown in Figure 2, and the embedding angle, sectional area, and element distance of rebar elements were defined. Literatures [24,25] obtained the material parameter values of cord by tensile test, which were used in this study, as listed in Table 2.

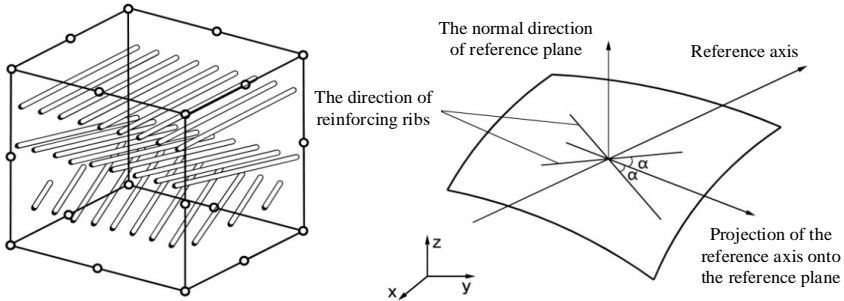

**Figure 2.** Schematic of embedded rebar elements.

**Table 2.** Parameter values of cord materials.

| Cord Materials | Density (kg/m³) | Angle with the Radial Surface of Tire (°) | Sectional Area (mm²) | Distance (mm) | Tensile Modulus (MPa) | Poisson's Ratio |
|---|---|---|---|---|---|---|
| Carcass | 1350 | 0 | 0.24 | 0.9 | 9597 | 0.29 |
| The first belt ply | 7800 | 63 | 0.21 | 1.4 | 195,351 | 0.29 |
| The second belt ply | 7800 | 117 | 0.21 | 1.4 | 195,351 | 0.29 |
| Cap ply | 1150 | 90 | 0.24 | 0.7 | 9907 | 0.30 |
| Tire bead | 7800 | 90 | 0.72 | 1.05 | 170,000 | 0.29 |

*2.2. Tire Modeling and Model Validation*

The modeling procedures of 3D-patterned tire started with a 2D half-section model, and then a 2D full-section tire model with mesh was obtained by mapping the model of central symmetry, as shown in Figure 3. The standard inflation pressure of 225/60R17 (0.25 MPa) was applied to the inner surface of the 2D tire model. A standard rim model was established by the analytical rigid body, and the rim model was assembled with the 2D tire model. The tire pattern model was established by a single tire pattern model, as shown in Figure 4. According to the size of single tire pattern, the 2D tire model was rotated by 4.9315° and assembled with the single tire pattern model to generate a tire segment. Seventy-three tire segments were generated using the same rotating method, and then the 3D patterned tire model was established. In the model, the rubber material was meshed as tetrahedron and hexahedron elements, and the element types were C3D8R and C3D6 in *ABAQUS*. The rebar element types of cord materials were SFM3D4R in *ABAQUS*. There are 273,513 elements and 357,987 nodes in the 3D-patterned tire model. The whole modeling flow and schematic diagram are shown in Figures 5 and 6.

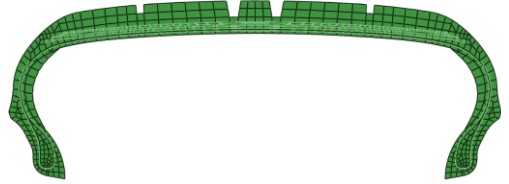

**Figure 3.** 2D tire model.

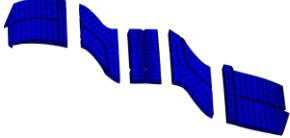

**Figure 4.** Single tire pattern model.

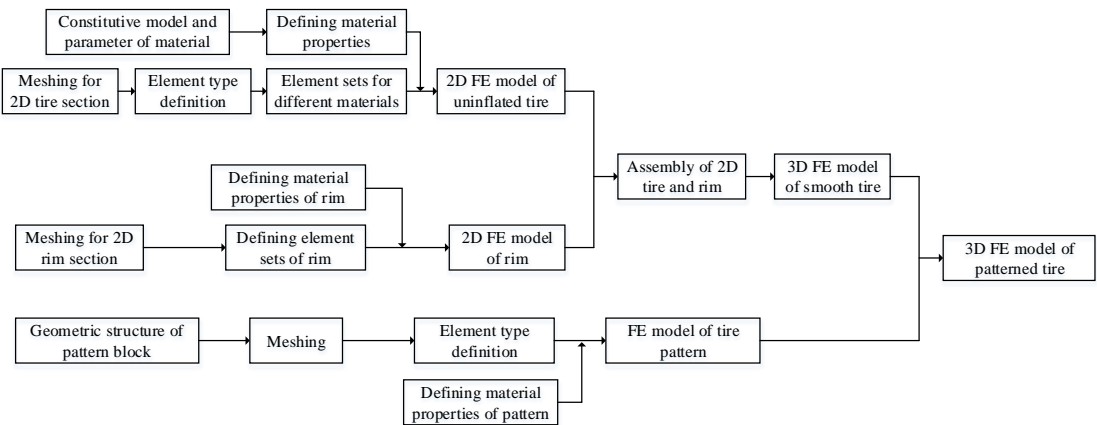

**Figure 5.** Modeling flow of 3D patterned tire.

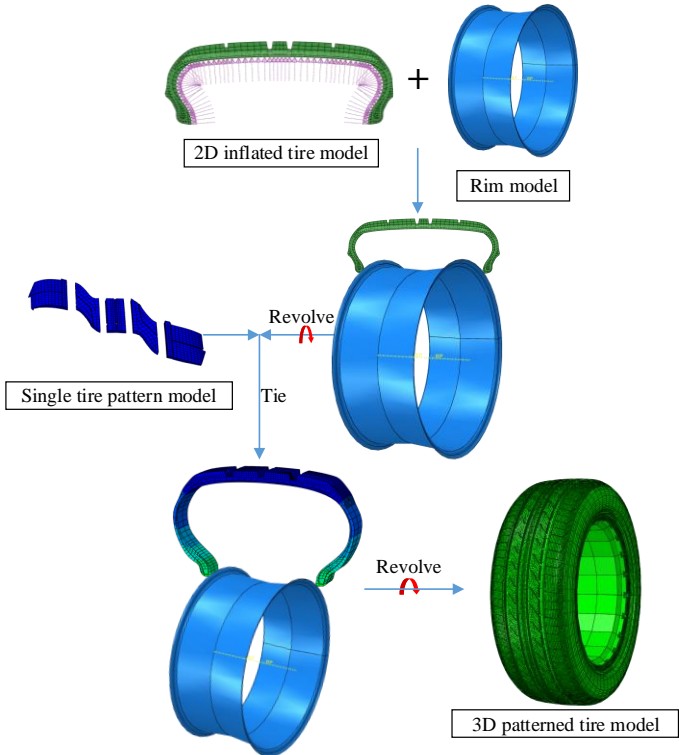

**Figure 6.** Modeling schematic of 3D patterned tire.

To validate the developed tire model, a tire stiffness test was conduct on the 225/60R17 tire by the Tire Dynamic Performance Tester (as shown in Figure 7a). Meanwhile, the tire stiffness test was simulated by the 3D patterned tire model, as shown in Figure 7b.

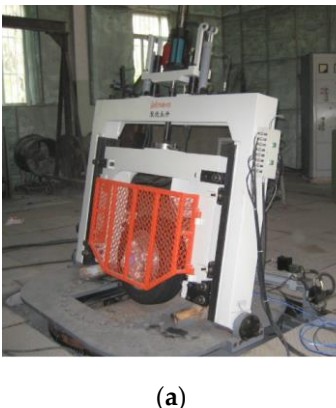
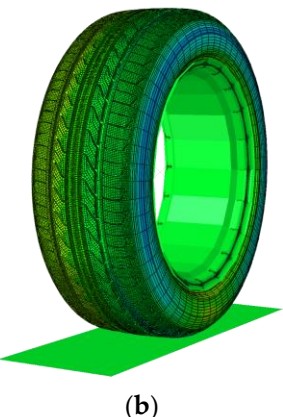

(**a**)　　　　　　　　　　　　　　　　　　　　(**b**)

**Figure 7.** Tire stiffness measurement and simulation tests (**a**) Tire Dynamic Performance Tester; (**b**) tire stiffness simulation model.

The measurement and simulation results are given in Figure 8. It can be observed that the measurement results and the simulation results are approximate to each other. Given the test error and modeling simplification of tire (such as the geometric structure and material characterization), it is regarded that the developed 3D-patterned tire model can simulate the mechanical properties of 225/60R17 tire effectively.

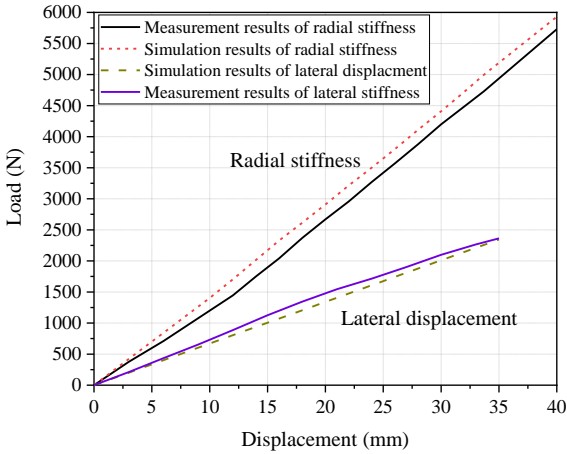

**Figure 8.** Comparison of tire stiffness test and simulation results.

### 2.3. Grounding Characteristics of Tire on SBDP

The grounding characteristics of tire have an effect on the hydroplaning behaviors on the SBDP. In this section, the tire–SBDP model was developed to study the grounding characteristics of static tire, which was used as the initial condition for tire rolling simulation in the following-up hydroplaning analysis. In the model, the augmented Lagrange method was used to define the contact between the tire and SBDP. According to the most unfavorable loading position for SBDP in the previous studies [20,25], the tire position on the SBDP was determined in the model. The SBDP model was composed of the orthotropic steel bridge deck (OSDB) and pavement. The OSBD model and pavement model were merged, and fix constraints were set to the bottom of OSBD. The pavement was modeled by the solid element (DC3D8 element in *ABAQUS*), and the OSBD was modeled by the shell element (DS3/DS4 in *ABAQUS*). There are 296534 elements in the tire–SBDP model, as shown in Figure 9, and some model parameters are listed in Table 3.

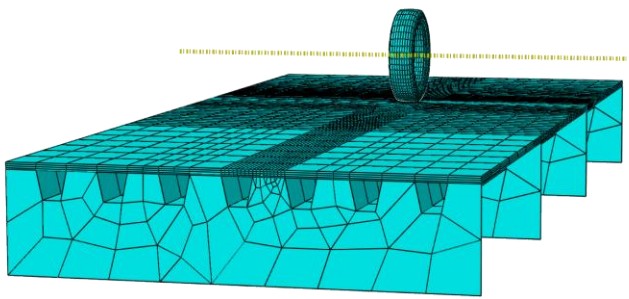

**Figure 9.** Tire–SBDP model.

The grounding characteristics of tire under different tire inflation pressures and wheel loads were calculated. Four tire inflation pressures of 0.2, 0.25, 0.29, and 0.35 MPa, and four wheel loads of 2.5, 3.0, 3.5, and 4.0 kN were taken for simulation [12–14,16]. Part of the tire tread imprint and grounding stress distribution calculation results are given in Figure 10.

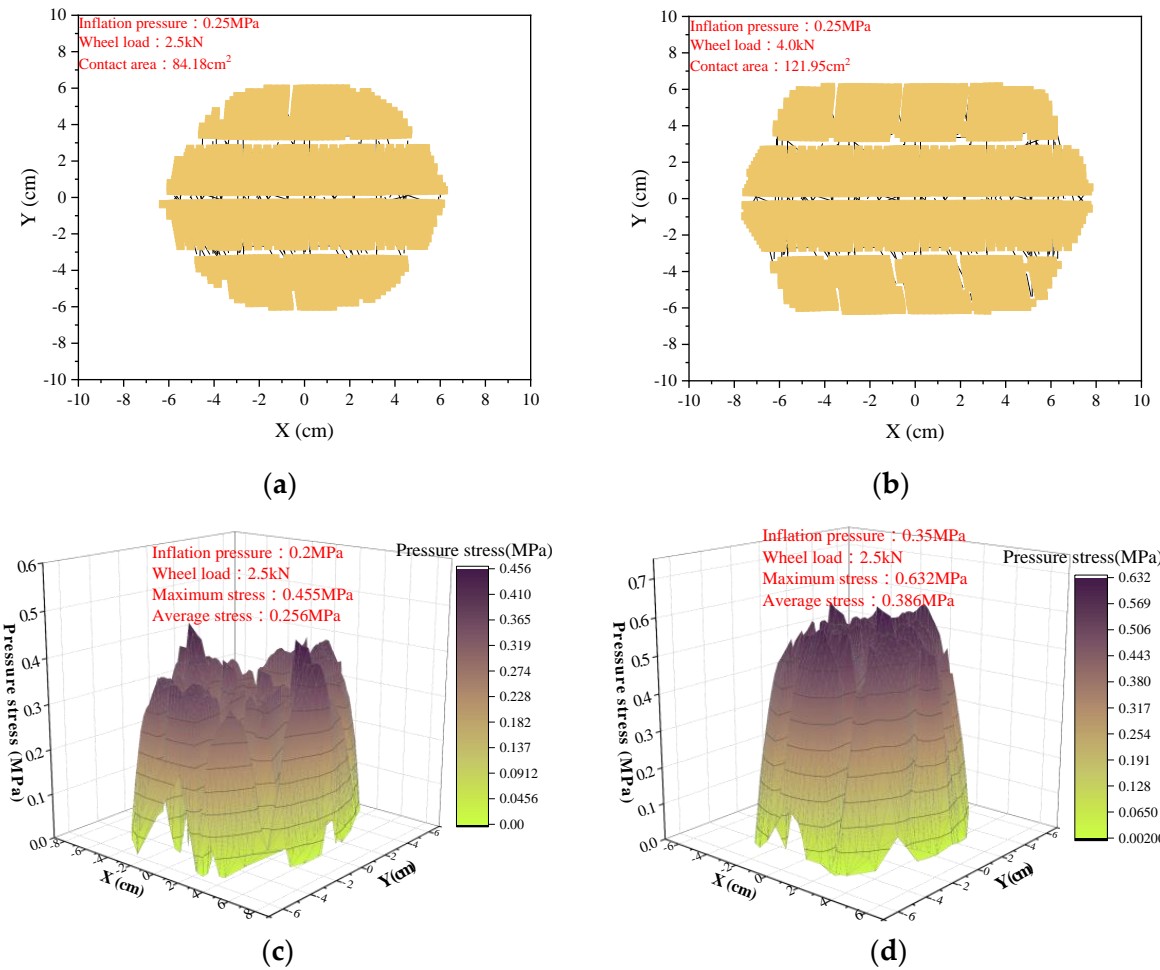

**Figure 10.** Tire tread imprints and grounding stress distributions on SBDP (**a**) tire tread imprint at 0.25 MPa inflation pressure and 2.5 kN wheel load; (**b**) tire tread imprint at 0.25 MPa inflation pressure and 4.0 kN wheel load; (**c**) grounding stress distribution at 0.2 MPa inflation pressure and 2.5 kN wheel load; (**d**) grounding stress distribution at 0.35 MPa inflation pressure and 2.5 kN wheel load.

**Table 3.** Parameters of tire–SBDP model.

| OSBD Length (m) | OSBD Width (m) | Deck Thickness (mm) | Pavement Thickness (mm) | Diaphragm Thickness (mm) | Diaphragm Distance (m) | | |
|---|---|---|---|---|---|---|---|
| 9.6 | 4.5 | 14 | 55 | 12 | 3.2 | | |
| | | **U-rib** | | | **Steel** | **Pavement** | |
| Top Width (mm) | Bottom Width (mm) | Height (mm) | Distance (mm) | Thickness (mm) | Modulus (MPa) | Poisson's Ratio | Modulus (MPa) | Poisson's Ratio |
| 300 | 180 | 280 | 380 | 6 | 210000 | 0.3 | 950 | 0.2 |

## 3. Tire–Fluid–Pavement Interaction Model

### 3.1. D Asphalt Pavement Model

The asphalt pavement for steel bridge was modeled based on the X-ray CT technology, and the modeling core of the pavement is the accurate reconstruction of pavement surface texture. The detailed procedures are summarized as below.

(1) Preparing the beam specimens (specimen size of 50 mm × 50 mm × 100 mm) of epoxy asphalt concrete (EAC) and SMA concrete, which are commonly used as the wearing surface materials of SBDP, as shown in Figure 11a. The aggregate gradations of EAC and SMA are given in Figure 12.

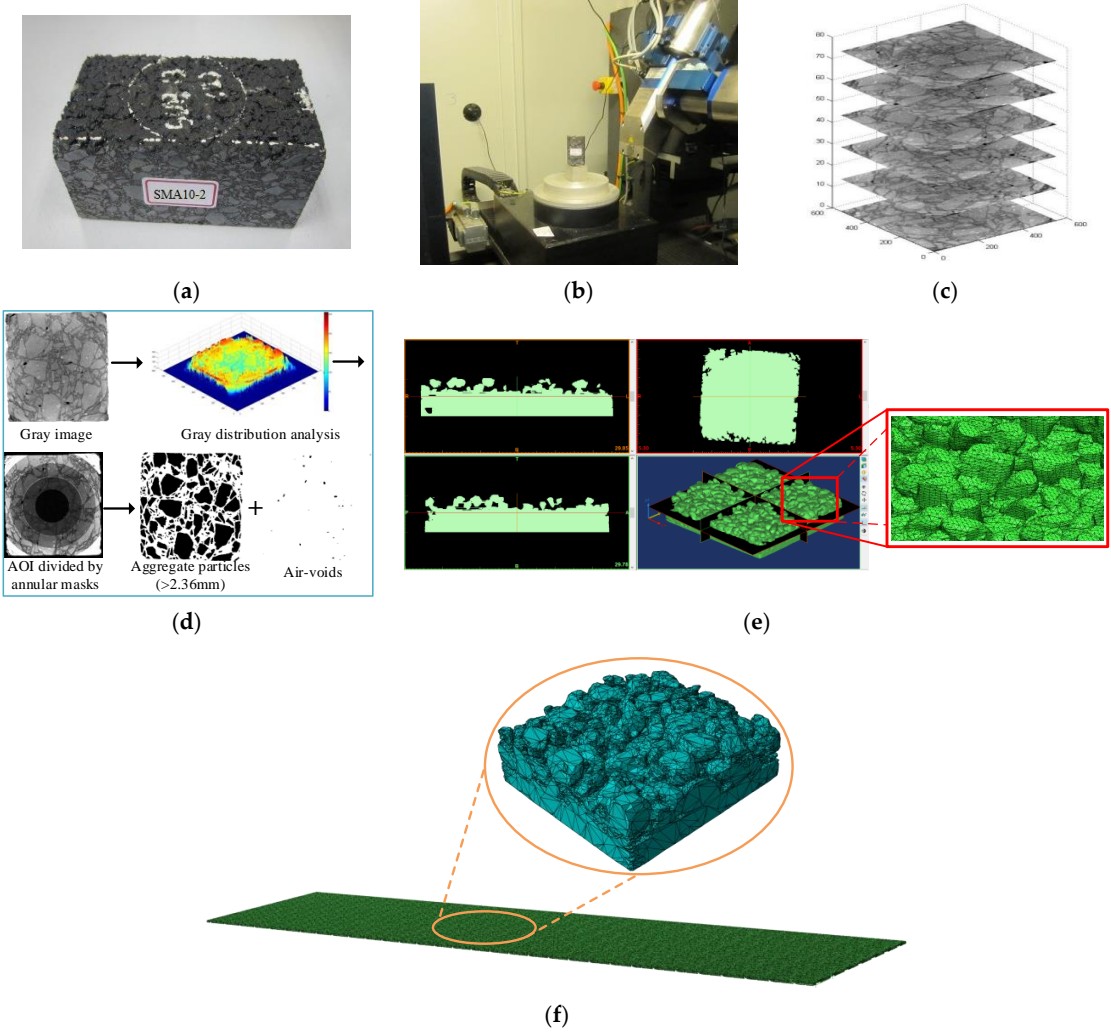

**Figure 11.** Modeling schematic of 3D asphalt pavement (**a**) beam specimens; (**b**) Y.CT Precision S model X-CT device; (**c**) tomographic images; (**d**) the image processing; (**e**) triangular element model of pavement; (**f**) 3D asphalt pavement model.

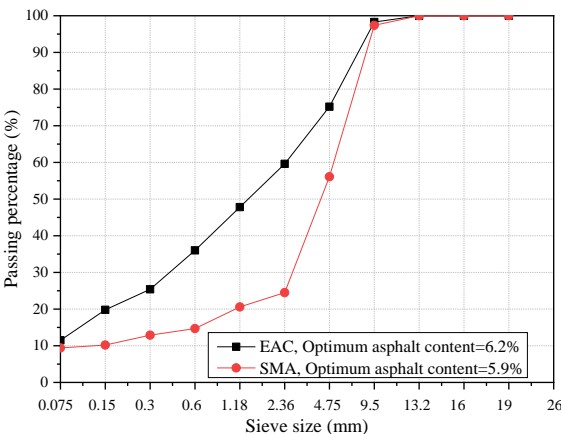

**Figure 12.** Aggregate gradations of EAC and SMA.

(2) The beam specimens were scanned by the X-ray CT technology. The X-ray CT device used in this study is a Y.CT Precision S model X-CT device, as shown in Figure 11b. The dimensions of the gray image were $1024 \times 1024$ pixel$^2$ and minimum resolution was 10 μm. The digital tomographic images were acquired at 0.1 mm intervals (Figure 11c), which assured the geometric characteristics of the aggregates and air-voids closely matched with the practical structure in the asphalt mixture.

(3) Image processing technology was used to obtain the precise morphology characteristics of SBDPs [26–29]. Annular masks with different radii were set in digital tomographic images through *MATLAB* software to divide the area of interest (AOI) into serial annular regions. After that, Otsu thresholding method was applied to each annular region to segment AOI from background. Finally, morphological operations (corrosion, filling) were implemented to process the initial binary images. The process is shown in Figure 11d.

(4) *Mimics* software was used to create the triangular element model based on a binary image sequence, and then *SolidWorks* software was used to convert the *stl* format of triangular element model into *x_b* format that can be identified by *ABAQUS* software, as shown in Figure 11e. The numerical models of aggregate particles and air-voids were imported into *ABAQUS*, Boolean operations, material property definitions, structure meshing operations were addressed to establish the 3D asphalt pavement model, as shown in Figure 11f.

### 3.2. Fluid Model

The fluid model was developed to simulate the dynamic characteristics of water film between the tire and SBDP. Firstly, the fluid geometric model was established by *Hypermesh* software, as shown in Figure 13a. Then, the geometric model was imported into the *ABAQUS* software to establish the fluid FE model by Euler elements, as shown in Figure 13b. The fluid model included the bottom water film region and the top air region, and the air region can ensure the enough space for fluid flow. The water film thickness is determined by the water film region size, and gravity acceleration of 9.8 m/s$^2$ was applied to the water film.

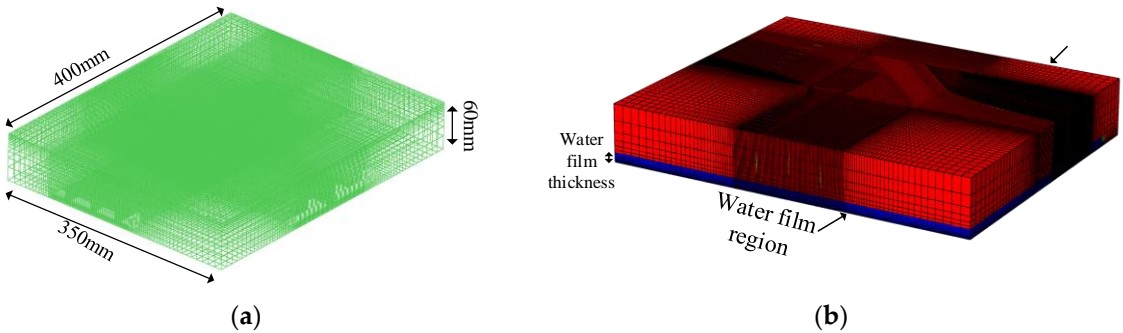

**Figure 13.** Fluid model (**a**) geometric model; (**b**) FE model.

The water film was regarded as Newtonian fluid in this study. The Mie–Grüneisen state equation was utilized to represent the relationship among fluid pressure, density, and unit mass internal energy as follows.

$$P = \frac{\rho_0 c_0^2 \eta}{(1 - s\eta)^2}(1 - \frac{\Gamma_0 \eta}{2}) + \Gamma_0 \rho_0 E_m \tag{4}$$

where $P$ is the pressure stress of fluid; $\rho_0$ is the initial density of fluid; $c_0$ is sound velocity in fluid at room temperature and pressure; $s$ and $\Gamma_0$ is the material parameter; $\eta = 1 - \rho_0/\rho$, $\rho$ is the fluid density after impact. The values of these parameters refer to the literature [15,30], and are listed in Table 4.

**Table 4.** Parameter values of Mie–Grüneisen state equation.

| Parameters | Values |
|---|---|
| $\rho_0/\text{kg·m}^{-3}$ | 1000 |
| $c_0/\text{m·s}^{-1}$ | 1484 |
| $S$ | 1.979 |
| $\Gamma_0$ | 0.11 |

### 3.3. Hydroplaning Analysis

The tire model, fluid model, and pavement model were combined for hydroplaning analysis based on the CEL algorithm [12–14,31]. The tire model and pavement model were assembled, and the contact conditions were defined by the tire–SBDP interaction. Then, the tire deformation and grounding analysis, tire braking/rolling/cornering analysis were conducted through steady state analysis of 3D tire rolling on the pavement. The stable stress–strain field at the given rolling speed was obtained for the subsequent transient analysis. After that, the fluid model was imported and the contact among tire, water film, and pavement was defined as general contact condition. The stable stress–strain field obtained in the previous step was transferred as the initial condition of hydroplaning analysis. The rolling tire on the wet pavement was simulated using a relative motion method. The same translation velocity was applied to the fluid and pavement model, and the corresponding angular velocity was applied to the tire model. Accordingly, a hydroplaning model with a complex tire pattern was developed, as shown in Figure 14a. In addition, a hydroplaning model with simple longitudinal pattern (Figure 14b) was developed to compare the hydroplaning behaviors with different tire patterns. The hydroplaning analysis procedures can be summarized in Figure 15.

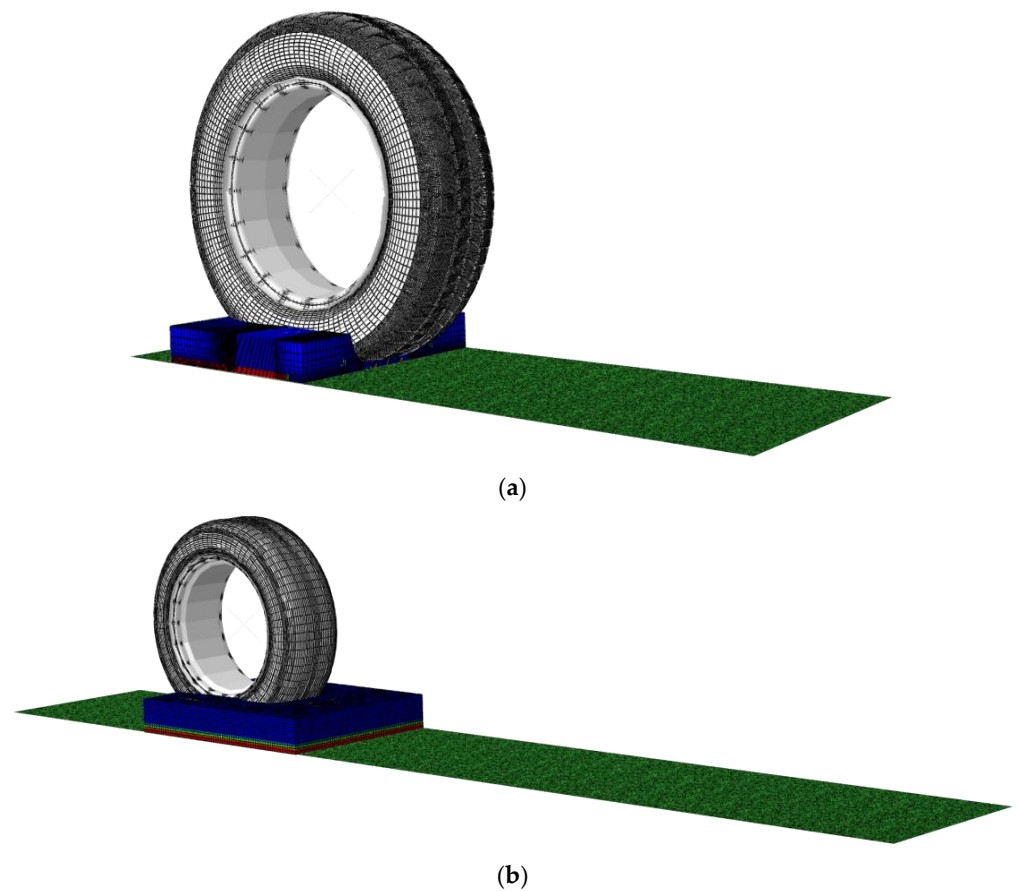

(**a**)

(**b**)

**Figure 14.** Hydroplaning model (**a**) complex tire pattern; (**b**) simple longitudinal tire pattern.

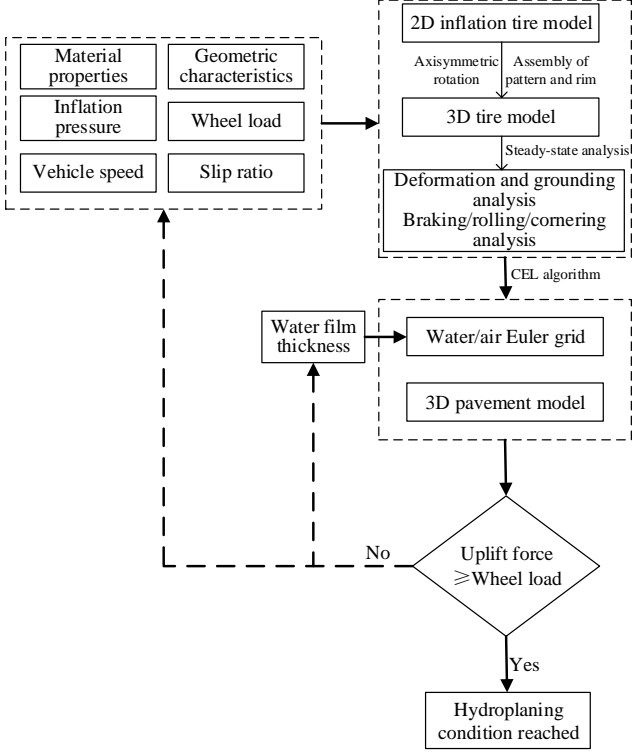

**Figure 15.** Hydroplaning analysis flow chart.

To validate the developed hydroplaning model, the modeling method was utilized to simulate the hydroplaning on the road pavement. The simulation results were compared with the calculation results by the NASA hydroplaning equation and Gallaway hydroplaning equation, as shown in Figure 16. As the NASA equation is applicable to hydroplaning assessment for smooth tires or tires with simple patterns, the hydroplaning speed simulation results of tires with longitudinal patterns and complex patterns are larger than the calculation results by NASA equation. The Gallaway equation is developed based on the hydroplaning test of complex patterned tires. It is observed that the simulation results of complex patterned tires and calculation results by Gallaway equation are approximate each other. Accordingly, it is regarded that the hydroplaning analysis method developed in this study is valid.

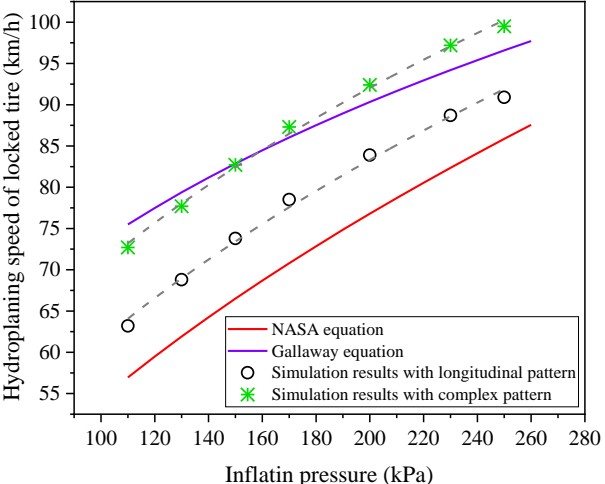

**Figure 16.** Results of numerical simulation and hydroplaning equations.

## 4. Hydroplaning Behaviors on SBDP

### 4.1. Fluid Imprint and Tire Force Analysis during the Hydroplaning

The fluid imprint variations when the tire is rolling over the wet pavement are shown in Figure 17. It is observed that the complex tire pattern can provide more drainage channels for water between tire and pavement, resulting in less water retained in the tire–pavement interface. This is why the hydroplaning speed of complex patterned tire is larger than that of longitudinal patterned tire. Therefore, a good tire pattern can effectively reduce the occurrence of hydroplaning.

When the tire is rolling on the wet pavement, the tire force diagram is shown in Figure 18. The hydrodynamic pressure increases with the vehicle speed. As a result, the water flow increases the uplift force and horizontal impact resistance on the tire. The increase in uplift force will reduce the contact area and friction force between tire and pavement. The hydroplaning occurs when the contact force between tire and pavement reduces to zero. Figure 19 gives the force variations on the tire with vehicle speed under the water film thickness of 10 mm. Due to the vibration response caused by structural damping in the tire during its motion, there are local oscillation changes in forces on the tire. However, the variation trends of uplift force, impact resistance, and friction force are consistent with the actual situation, which further verifies the validity of the developed hydroplaning model in this study. The hydroplaning speed corresponds to the vehicle speed when the friction force reduces to zero.

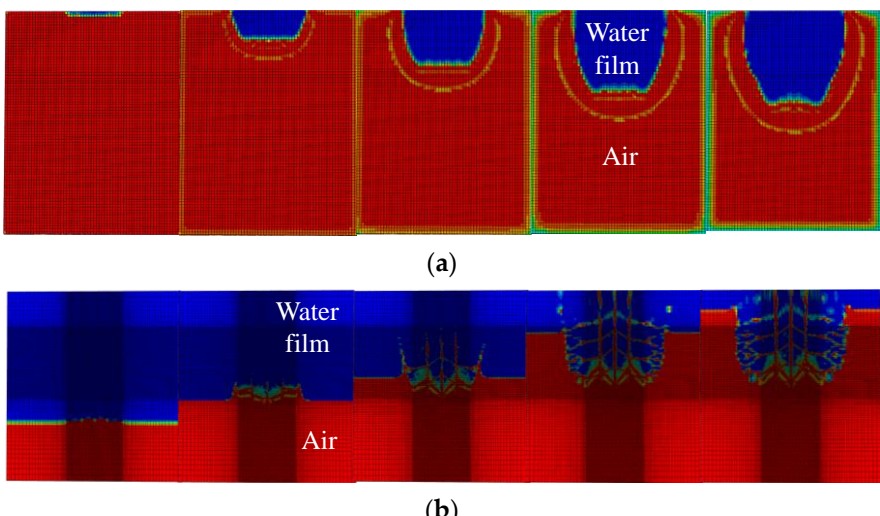

**Figure 17.** Fluid imprint variations when tire rolling on the wet pavement (**a**) longitudinal tire pattern; (**b**) complex tire pattern.

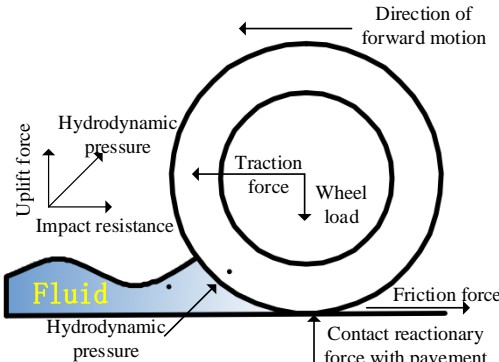

**Figure 18.** Force diagram of tire during the hydroplaning.

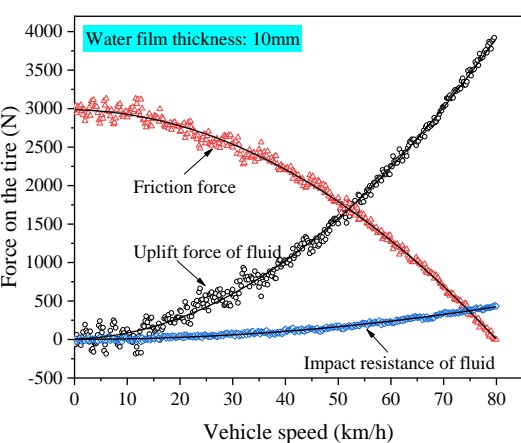

**Figure 19.** Force variations on the tire with vehicle speed.

### 4.2. Impact Analysis of Hydroplaning

The hydroplaning analysis on the SBDP under different conditions was conducted, and the results are given in Figure 20. The hydroplaning speed decreased with the increase in tire slip ratio, and the hydroplaning speed decreased by about 8% when the slip ratio increased from 20 to 100%. This is because water flow is more easily immersed in the tire–pavement interface when the tire slip speed (or slip ratio) increases. Accordingly, it is

also a good explanation that emergency braking should be avoided when driving in rainy days, but should slow down until the vehicle stop. The hydroplaning speed also decreases with the increase in tire inflation pressure because that the tire slip speed increases under lower inflation pressure. Therefore, the tire should have adequate inflation pressure when driving on rainy days.

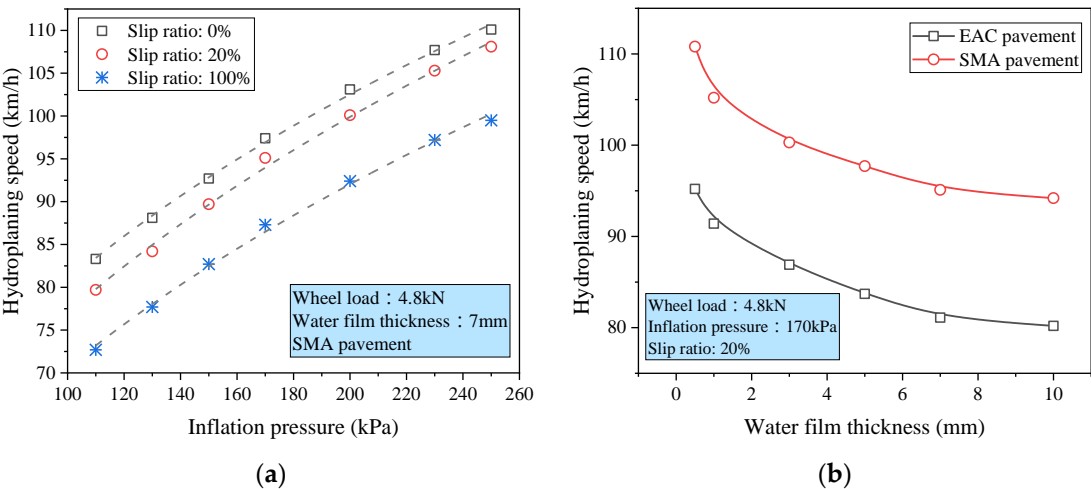

**Figure 20.** Hydroplaning results under different conditions (**a**) hydroplaning speed variations with inflation pressure under different slip ratios; (**b**) hydroplaning speed variations with water film thickness under different SBDPs.

As shown in Figure 20b, the hydroplaning speed decreases with the increase in water film thickness. Meanwhile, it can be seen from the variation trend that the descent of hydroplaning speed is rapid when the water film thickness is small, but when the water film thickness exceeds 7 mm, the descent of hydroplaning speed is not significant. Due to the better macrotexture of SMA pavement, the hydroplaning speed on the SMA pavement is remarkably higher than that on the EAC pavement. The contribution of pavement macrotexture is similar to that of tire pattern, which is to provide drainage channel for water. Therefore, the aggregate gradation optimization for SBDP is vital to reduce the occurrence of hydroplaning.

## 5. Conclusions

This study developed a tire–fluid–pavement interaction model for hydroplaning analysis on the SBDP. The results of this study can be summarized as follows:

(1) The developed tire–fluid–pavement interaction model for SBDP based on the CEL algorithm was verified by the NASA hydroplaning equation and the Gallaway hydroplaning equation. The model can be used to evaluate the hydroplaning performance for tire on the SBDP.

(2) The complex tire pattern can provide more drainage channels for water and reduce the occurrence of hydroplaning.

(3) The hydroplaning speed decreases with the increase in the tire slip ratio, and increases with the tire inflation pressure. The hydroplaning speed decreases by about 8% when the slip ratio increases from 20 to 100%, so emergency braking should not be performed when driving on rainy days, but drivers should slow down until the vehicle stops. At the same time, the tire should have adequate inflation pressure to improve the anti-hydroplaning performance in rainy days.

(4) The hydroplaning speed decreases with the increase in water film thickness. The descent of hydroplaning speed is rapid when the water film thickness is small, but when the water film thickness exceeds 7 mm, the descent of hydroplaning speed is not significant.

(5) The macrotextures on the SBDP surface can form drainage channels for water, and the aggregate gradation optimization for SBDP is vital to reduce the occurrence of hydroplaning.

**Author Contributions:** Conceptualization, Y.L. and Z.Q.; methodology, Y.L. and C.L.; investigation, Y.L. and Q.H.; data curation, C.L. and Q.H.; software, Y.L.; writing-original draft, Y.L.; writing—review and editing, Z.Q. and Q.H.; funding acquisition, Y.L. All authors have read and agreed to the published version of the manuscript.

**Funding:** This research was funded by the National Natural Science Foundation of China, grant number 52008102 and the Natural Science Foundation of Jiangsu Province, grant number BK20200384.

**Institutional Review Board Statement:** Not applicable.

**Informed Consent Statement:** Not applicable.

**Data Availability Statement:** Data are available upon request from authors.

**Conflicts of Interest:** The authors declare no conflict of interest.

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
