# Peer review of "Investigation on Hydroplaning Behaviors of a Patterned Tire on a Steel Bridge Deck Pavement"

_applsci, doi:10.3390/app112210566_

Round 1
Reviewer 1 Report
Numerical investigation on the hydroplaning behaviours of patterned tire has been performed, calibrating the numerical model with empirical formulas.
The paper is written in a very clear way, with an acceptable English language. However, some points have to be better explained prior to accept the paper. Here are reported some considerations:
- Abstract, line 18. Please, report the meaning of CEL, i.e. Couple Eulerian-Lagrangian algortihm.
- Figure 1b. What do the reported lines represents? Are they experimental or Yeoh model fitted data? Please, clarify this point.
- Section 2.2 Tire modelling. Please, reports more FE model information. For example, the adopted software, number and type of the adopted elements.
- Section 2.3 Grounding Characteristics of Tire. Please, reports more information regarding the contact modelling, i.e. number and type of elements.
- Section 2.3 Grounding Characteristics of Tire, line 165-167. Why the Authors adopted these particular values of tire inflation pressure and wheel loads. Where were they taken from?
- Figure 19. Are the black lines of the figure obtained by data fittings?
For all the previous reasons, the reviewer recommends minor amendments of paper for publication in Applied Sciences.
Author Response
Point 1: Abstract, line 18. Please, report the meaning of CEL, i.e. Couple Eulerian-Lagrangian algortihm.
Response 1: Thanks for the comment of the reviewer. The authors have added the meaning of CEL in the Abstract.
Point 2: Figure 1b. What do the reported lines represents? Are they experimental or Yeoh model fitted data? Please, clarify this point.
Response 2: The reported lines in Figure 1b are the experimental data.
Point 3: Section 2.2 Tire modelling. Please, reports more FE model information. For example, the adopted software, number and type of the adopted elements.
Response 3: Thanks for the comment of the reviewer. The authors have added more information on the FE model in the revised manuscript.
Point 4: Section 2.3 Grounding Characteristics of Tire. Please, reports more information regarding the contact modelling, i.e. number and type of elements.
Response 4: Thanks for the comment of the reviewer. The authors have added more information on the tire-SBDP model in Section 2.3.
Point 5: Section 2.3 Grounding Characteristics of Tire, line 165-167. Why the Authors adopted these particular values of tire inflation pressure and wheel loads. Where were they taken from?
Response 5: For the 225/60R17 radial tire, the standard tire inflation pressure is 0.25MPa. Given the situations of insufficient tire inflation pressure and excessive tire inflation pressure, four tire inflation pressures of 0.2MPa, 0.25MPa, 0.29MPa, and 0.35MPa were selected. While the four wheel loads of 2.5kN, 3.0kN, 3.5kN, and 4.0kN were taken according to the references [12-14]. The authors have added the citation of references for the selection of these particular values.
Point 6: Figure 19. Are the black lines of the figure obtained by data fittings?
Response 6: Yes, the black lines were obtained by data fittings.
Reviewer 2 Report
- It is good and recommended for the authors to cite/refer between 30-40 references.
- The Conclusion section to be written in paragraph forms.
- It is recommended for the authors to compare and discuss their findings with previous studies done by other researchers. Please revise the Results and Discussion section accordingly.
- 20 figures for one paper is quite extensive. It is highly recommended for the authors to choose the most relevant one, if possible to reduce between 10-12 figures only.
- Overall, this is an interesting research and the paper is well described. Very well done.
Author Response
Point 1: It is good and recommended for the authors to cite/refer between 30-40 references.
Response 1: Thanks for the comment of the reviewer. The authors have cited more references in the revised manuscript.
Point 2: The Conclusion section to be written in paragraph forms.
Response 2: The Conclusion section is written in paragraph forms.
Point 3: It is recommended for the authors to compare and discuss their findings with previous studies done by other researchers. Please revise the Results and Discussion section accordingly.
Response 3: Thanks for the comment of the reviewer. The hydroplaning analysis for the steel bridge deck pavement had never been made before this study. The previous studies were related to the hydroplaning on the ordinary road pavement. The authors can make a comparison of the hydroplaning characteristics on the steel bridge deck pavement and ordinary road pavement in the future study, to explain why steel bridge deck pavement is more prone to traffic accidents.
Point 4: 20 figures for one paper is quite extensive. It is highly recommended for the authors to choose the most relevant one, if possible to reduce between 10-12 figures only.
Response 4: Thanks for the comment of the reviewer. As the modeling process of the tire-fluid-pavement for the steel bridge deck is a complex and massive work, and these figures help to introduce the work. Therefore, the authors think it is better to retain these figures.
Point 5: Overall, this is an interesting research and the paper is well described. Very well done.
Response 5: Thanks very much for the reviewer’s comments and suggestions.
Reviewer 3 Report
Manuscript Number: applsci-1451286
Title: Investigation on hydroplaning behaviors of patterned tire on the steel bridge deck pavement
In this manuscript, a numerical approach was put forward to predict the hydroplaning behaviors of tires on the steel bridge deck pavement. Several issues should be properly addressed.
- The authors should be noticed that references are always lacked in statements require evidences. For example, in Line 25-35 (i.e. the first paragraph in introduction) where hydroplaning and hydroplaning speed are defined and introduced, no reference is provided. Are these innovative concepts? In Line 79, it is stated that “survey shows that the hydroplaning propensity on SBDP is higher than road pavement”, and where is the survey? Please at least provide a reference.
- Line 30, “generating” should be “generate”.
- Line 79, “survey” should be “a survey”.
- Line 86, “it should to” is a grammatical mistake.
- Line 88. “complex macrotexture” seems to be an innovative point in this research, but its significance was not mentioned in the context.
- Line 92, why this type of tire is used?
- The test specimens in Figure 1a seem to be prepared according to certain standard, please provide the reference if possible.
- Line 121-122, why this modelling procedure is used instead of building the whole-tire model all at once.
- Line 126, a reminder is that please make sure you have permission to use the “single tire pattern model provided by a tire manufacturer” in publication.
- Line 214, why the water film can be regarded as Newtonian fluid.
- Line 244. The effectiveness of the modelling was verified by the NASA and Gallaway models. Then, what is the value of this current work is the two models are effective enough to predict this behavior. Is it possible to compare these results with existing experimental results?
- Line 284. Please provide a definition for “tire slip ratio”.
- A basic mistake exists in Figure 18. A hint is that, there are altogether two horizontal forces shown in Figure 18, i.e. the impact resistance caused by hydro pressure and the friction force; they both push the tire to the right hand side, then how can the forces be balanced?

Author Response
Point 1: The authors should be noticed that references are always lacked in statements require evidences. For example, in Line 25-35 (i.e. the first paragraph in introduction) where hydroplaning and hydroplaning speed are defined and introduced, no reference is provided. Are these innovative concepts? In Line 79, it is stated that “survey shows that the hydroplaning propensity on SBDP is higher than road pavement”, and where is the survey? Please at least provide a reference.
Response 1: Thanks for the comment of the reviewer. The concepts of hydroplaning and hydroplaning speed are not innovative. The hydroplaning is a common phenomenon on the wet pavement, and the authors have explained the hydroplaning phenomenon and the hydroplaning speed. The authors have provided the reference about hydroplaning in the revised manuscript. Regarding the statement of “the hydroplaning propensity on SBDP is higher than road pavement”, the statement is not properly. The authors have revised as “The hydroplaning propensity on SBDP is generally higher than road pavement due to the small pavement macrotexture depth”, and the references are provided.
Point 2: Line 30, “generating” should be “generate”.
Response 2: Thanks for the comment of the reviewer. The authors have revised it in the revised manuscript.
Point 3: Line 79, “survey” should be “a survey”.
Response 3: Thanks for the comment of reviewer. This sentence has been revised.
Point 4: Line 86, “it should to” is a grammatical mistake.
Response 4: Thanks for the comment of the reviewer. The authors have revised this sentence.
Point 5: Line 88. “complex macrotexture” seems to be an innovative point in this research, but its significance was not mentioned in the context.
Response 5: In Line 79, the influence of macrotexture on the hydroplaning was mentioned in the revised manuscript (i.e. the hydroplaning propensity on SBDP is generally higher than road pavement due to the small pavement macrotexture depth). The modeling core of pavement in this study is the accurate reconstruction of pavement surface macrotexture. In addition, the macrotexture levels of EAC pavement and SMA pavement are different. This study compared the hydroplaning behaviors on the EAC pavement and the SMA pavement in Figure. 20b, to reveal the influence of pavement macrotexture on the hydroplaning.
Point 6: Line 92, why this type of tire is used?
Response 6: The 225/60R17 radial tire was used in this study because that it is a common tire type for vehicles. For different types of tires, the rubber materials are different, resulting in the different material properties for tire components. One of the keys to tire modeling is the material parameter acquisitions, while the material parameter acquisitions should obtain the raw tread rubber, cap ply rubber, carcass rubber, etc, firstly. These raw rubber materials of 225/60R17 had been obtained from a tire factory in advance. Therefore, the 225/60R17 radial tire was used in this study. By the way, the authors happened to have a spare 225/60R17 can be used for tire stiffness test.
Point 7: The test specimens in Figure 1a seem to be prepared according to certain standard, please provide the reference if possible.
Response 7: Yes, the dumbbell-shape specimens were prepared and the uniaxial tensile test was conducted according to the standard, ASTM D638. The authors have provided the reference in the revised manuscript.
Point 8: Line 121-122, why this modelling procedure is used instead of building the whole-tire model all at once.
Response 8: The building of the whole tire model at once is difficult to achieve, especially for the accurate assembly of smooth tire, tire pattern, and the rim. The modeling procedure based on the revolve function in ABAQUS is the general modeling method for tire, and the modeling procedure in this study can ensure the accurate assembly of tire components.
Point 9: Line 126, a reminder is that please make sure you have permission to use the “single tire pattern model provided by a tire manufacturer” in publication.
Response 9: Thanks for the reminder of the reviewer. The authors make sure that the use of the single tire pattern model will be no commercial and legal disputes. For protection purposes, the information of tire manufacturer was not provided. The authors have revised this sentence to avoid bringing distraction to readers.
Point 10: Line 214, why the water film can be regarded as Newtonian fluid.
Response 10: Many fluids in nature are Newtonian fluids. Most pure liquids such as water and alcohol, light oil, low molecular compound solutions and low-velocity flowing gases are Newtonian fluids. Polymers concentrated solutions and suspensions of polymers are generally non-Newtonian fluids. As a Newtonian fluid, the water film obeys the conservation of mass, momentum, and energy. Accordingly, the Mie-Grüneisen state equation can be utilized.
Point 11: Line 244. The effectiveness of the modelling was verified by the NASA and Gallaway models. Then, what is the value of this current work is the two models are effective enough to predict this behavior. Is it possible to compare these results with existing experimental results?
Response 11: As stated in the Introduction section, the hydroplaning test needs demanding site and equipment, and it has high test cost and potential safety hazard. There is no available experimental results on the steel bridge deck pavement to compare with the simulation results. Therefore, the authors verified the modeling method in this study by NASA and Gallaway model through simulating the NASA and Gallaway hydroplaning tests (the tests were conducted on the road pavement). However, the NASA and Gallaway model are not enough to support this study, such as the force analysis of tire during the hydroplaning. In addition, the NASA and Gallaway model for road pavements are not necessarily applicable to the steel bridge deck pavement.
Point 12: Line 284. Please provide a definition for “tire slip ratio”.
Response 12: Thanks for the comment of the reviewer. The authors have provided the definition of slip ratio in the Equation (2) of the revised manuscript.
Point 13: A basic mistake exists in Figure 18. A hint is that, there are altogether two horizontal forces shown in Figure 18, i.e. the impact resistance caused by hydro pressure and the friction force; they both push the tire to the right hand side, then how can the forces be balanced?
Response 13: Thanks for the comment of the reviewer. The traction force of vehicle to the tire was ignored, and the authors have revised it in Figure 18.
Reviewer 4 Report
I have a few comments that might be usefully addressed to improve the overall quality of the paper and some minor observation:
- The introduction should be more detailed with some other papers result in this field. Some of the papers are already cited, but the introduction should contain a little bit more presentation of other reaserchers works.
- The main content of the paper is very well written, but the conclusions need improvement, now they are vaguely written text in order to highlight the results obtained.
- The experimental program is detailed and the quality of the results’ discussion in its current form satisfies the requirements necessary for a research paper.
The main problem of the article is the comparision between two different asphalt mixures, with diferent macrotextures, and the results are very much influenced by this difference.
My opinion is to try to explain better the effect of asphalt mixture macrotexture in hydroplaning phenomenon.
The paper is therefore very well suited to this journal. The authors are to be commended on the professional quality of the research and the paper.
Author Response
Point 1: The introduction should be more detailed with some other papers result in this field. Some of the papers are already cited, but the introduction should contain a little bit more presentation of other researchers’ works.
Response 1: Thanks for the comment of the reviewer. The authors have added more presentation of other papers’ results in this field.
Point 2: The main content of the paper is very well written, but the conclusions need improvement, now they are vaguely written text in order to highlight the results obtained.
Response 2: Thanks for the comment of the reviewer. The authors have revised the conclusions according to the suggestion of the reviewer.
Point 3: The experimental program is detailed and the quality of the results’ discussion in its current form satisfies the requirements necessary for a research paper.
Response 3: Thanks for the comment of the reviewer. The authors optimize this paper as possible.
Point 4: The main problem of the article is the comparison between two different asphalt mixtures, with different macrotextures, and the results are very much influenced by this difference. My opinion is to try to explain better the effect of asphalt mixture macrotexture in hydroplaning phenomenon.
Response 4: Thanks for the comment of the reviewer. The macrotexture has great influence on the hydroplaning of asphalt pavement according to the other researchers’ previous works. Only two asphalt mixtures (SMA and EAC) were studied in this paper, the data is insufficient to explain the effect of macrotexture on the hydroplaning phenomenon. The authors are carrying out the study on the effect of asphalt mixture macrotexture on the hydroplaning and will present the results in another paper. In the current study, five types of asphalt mixtures which are applicable for steel bridge deck pavement are taken for hydroplaning analysis, and the macrotexture levels of these mixtures are measured to reveal the effect of macrotexture on the hydroplaning.
Reviewer 5 Report
The subject of the research is the cooperation between the tire and the road surface. The manuscript is an interesting presentation of the research conducted and the results obtained as well. The article is well structured and fits thematically to the journal. However, minor corrections should be made before publication:
- in the introduction there are relatively few citations and much of it is more than ten years old, the reviewer recommends enriching the section with newer literature items.
- conclusions should be expanded
Author Response
Point 1: In the introduction there are relatively few citations and much of it is more than ten years old, the reviewer recommends enriching the section with newer literature items.
Response 1: Thanks for the comment of the reviewer. The authors have revised the introduction section according to the suggestion of the reviewer, including that the research results in this field by other researchers in recent years are provided.
Point 2: Conclusions should be expanded
Response 2: Thanks for the comment of the reviewer. The authors have expanded the conclusions in the revised manuscript, and some quantitative conclusions are added.
Round 2
Reviewer 3 Report
The manuscript has been well revised and can be accepted.